# Age, Sex and BMI Relations with Anti-SARS-CoV-2-Spike IgG Antibodies after BNT162b2 COVID-19 Vaccine in Health Care Workers in Northern Greece

**DOI:** 10.3390/microorganisms11051279

**Published:** 2023-05-13

**Authors:** Paraskevi Papaioannidou, Kalypso Skoumpa, Christos Bostanitis, Maria Michailidou, Theodouli Stergiopoulou, Ioannis Bostanitis, Maria Tsalidou

**Affiliations:** 11st Department of Pharmacology, School of Medicine, Faculty of Health Sciences, Aristotle University of Thessaloniki, 541 24 Thessaloniki, Greece; 2Microbiological Department, General Hospital of Katerini, 601 00 Katerini, Greece

**Keywords:** anti-SARS-CoV-2-Spike antibodies, SARS-CoV-2 IgG, BNT162b2 COVID-19 vaccine, COVID-19 immunization, age, sex, BMI, body mass index, obesity, Greece

## Abstract

The aim of this work was to study age, sex, and BMI (Body Mass Index)-related differences in the development of anti-SARS-CoV-2-Spike IgG antibodies, after vaccination with the BNT162b2 COVID-19 vaccine, in health care workers of a General Hospital in a city in Northern Greece. Blood sampling was drawn two to four weeks following the second dose of the vaccine, and six months after the first blood sample collection. Measurement of serum IgG antibodies against the spike domain of SARS-CoV-2 was performed using the SARS-CoV-2 IgG II Quant assay. All participants had sufficient serum IgG titers in the first measurement. Women developed higher IgG titers than men. The IgG titers were inversely related to age in both sexes; there was also a small, insignificant tendency to be inversely related to BMI. Six months after the first measurement, the IgG titers decreased dramatically to values less than 5% of the initial. This decrease was observed in both men and women and was inversely related to age. Multivariate regression analysis showed that age and sex explained with statistical significance 9% of the variance in SARS-CoV-2 IgG titers in our study population; the role of BMI was limited and insignificant.

## 1. Introduction

Age, sex, and obesity have been among the main risk factors for morbidity and mortality due to COVID-19. Although in the beginning of the pandemic, the emphasis was placed on old age and pre-existing health problems such as obesity, hypertension, and diabetes [1,2,3,4,5], it was soon realized that male sex was also a serious risk factor [6,7]. The epidemiological findings from all over the world showed higher morbidity and mortality in males than in females [8]. This finding was attributed to sex-based immunological differences and to higher expression of angiotensin-converting enzyme-2 receptors in males than in females, as well as to differences in gender lifestyle, such as levels of smoking and drinking, and different attitudes in preventive measures against SARS-CoV-2 infection, such as handwashing, mask wearing, and staying at home [8,9,10].

In spite of the numerous studies on the effect of age, sex, and obesity in COVID-19 morbidity and mortality, the effect of age, sex, and obesity on immunization after COVID-19 vaccination has not been studied extensively, and the relative studies, especially from Northern Greece, are rather limited. Apart from that, in most cases, researchers have used different vaccines, different study protocols, and different time points after vaccination for their studies.

The aim of this prospective observational study was to investigate the relation of age, sex, and Body Mass Index (BMI) to the development of anti-SARS-CoV-2-Spike IgG antibodies after vaccination with the BNT162b2 COVID-19 vaccine in health care workers of a general hospital of a small city in Northern Greece, at two different time points (two to four weeks following the second dose of the vaccine, and six months after the first measurement).

## 2. Materials and Methods

### 2.1. Study Design

The participants of this prospective observational study were health care workers of the General Hospital of Katerini, Greece, who had been vaccinated with two doses of the Pfizer/BioNTech BNT162b2 COVID-19 vaccine (Comirnaty), 21 days apart according to recommendations, in January and February 2021. The participants belonged to the following professional categories: medical doctors, biologists, nurses, social health workers, auxiliary health workers and administrative staff members. All participants received the same type of COVID-19 vaccine (Comirnaty), that is a monovalent mRNA vaccine that encodes the spike protein of only the original SARS-CoV-2. At the time of the initiation of vaccination at the General Hospital of Katerini, the monovalent BNT162b2 COVID-19 vaccine was the only available COVID-19 vaccine in Greece.

328 participants were registered in the study that started in February 2021 and ended in August 2021. The protocol of the study was approved by the Ethics Committee of the General Hospital of Katerini (protocol code 4.19/2021/4226–2021). Written informed consent was obtained from all participants involved in the study, and the data were processed according to the institutional and national ethical standards and the World Medical Association (WMA) Declaration of Helsinki (1975), as revised in 2013.

Among the 328 participants that were enrolled in the study, 33 subjects reported previous COVID-19 infection approximately 2–3 months before vaccination and 5 subjects did not receive the second dose of the vaccine due to COVID-19 disease or allergic reaction to the first dose. These subjects were excluded from the study. Thus, finally, 290 participants were included in the study.

Blood sample collections were performed at two time points: (1) two to four weeks following the second dose of the vaccine, and (2) six months after the first blood sample was drawn. The schedule of vaccination and measurement of serum IgG antibodies against the spike domain of SARS-CoV-2 was as follows:D1 (Dose 1): the day of the administration of the BNT162b2 dose 1 (5 January 2021 to 5 February 2021).D2 (Dose 2): the day of the administration of the BNT162b2 dose 2 (3 weeks after D1, 26 January 2021 to 26 February 2021).M1 (Measurement 1): the day of the 1^st^ blood sample collection (2–4 weeks after D2, 9 February 2021 to 9 March 2021).M2 (Measurement 2): the day of the 2nd blood sample collection (6 months after M1, 9 August 2021 to 31 August 2021).

In the first step of the study (S1, Study 1), we analyzed the antibody response of the 1st blood sample collection (M1) that was drawn 2–4 weeks after D2 (9 February 2021 to 9 March 2021, *n* = 290).

In the second part of the study (S2, Study 2), we analyzed the antibody response of the 2nd blood sample collection (M2) that was drawn 6 months after M1 (9 August 2021 to 31 August 2021, *n* = 180). The participants of the second part of the study were selected at random among the 290 subjects that finally participated in the first part, according to the order of appearance. Participants infected by SARS-CoV-2 in the time period from M1 to M2 were excluded from the study S2.

### 2.2. Specimen Collection and Biobanking

Blood samples were collected in the Department of Microbiology, General Hospital of Katerini, and were prepared using standard operating procedures.

The samples were collected using a winged drawing system, equipped with a 21-gauge needle (BD Vacutainer) connected to a needle holder (BD Vacutainer). For serum preparation, the blood was stored in clot activator tubes (CAT, BD Vacutainer) and left for at least 30 min to completely coagulate, according to the manufacturer’s instructions. The tubes were then centrifuged at 1500 RCF for 10 min at room temperature, and the sera were separated manually from the blood clot and aliquoted in cryovials with a one-dimensional barcode. The aliquots were then kept at −20 °C until immunological analysis. 

### 2.3. Quantitative Analysis of Anti-Trimeric Spike IgG Antibodies

Measurement of serum IgG antibodies against the spike domain of SARS-CoV-2 was performed using the SARS-CoV-2 IgG II Quant assay, a chemiluminescent microparticle immunoassay (CMIA) provided by Abbot Diagnostics [11,12,13] that is used for the qualitative and quantitative determination of the IgG antibodies against the spike receptor-binding domain (RBD) of SARS-CoV-2.

The SARS-CoV-2 IgG II Quant assay is a chemiluminescent microparticle immunoassay (CMIA) used for the qualitative and quantitative determination of IgG antibodies to SARS-CoV-2 in human serum and plasma on the Alinity and ARCHITECT i Systems. The SARS-CoV-2 IgG II Quant assay may be used as an aid for evaluating immunological status after vaccination and for the diagnosis of SARS-CoV-2 infection, in conjunction with clinical presentation and other laboratory tests.

As reported by the manufacturer’s instructions, the assay range is 0–40,000 AU/mL. A positive result (>50 AU/mL) indicates the presence of antibodies following vaccination and/or infection. The normal range of positive results is 50–25,000 AU/mL. According to published research from the University of Washington, this test has 99.9% specificity and 100% sensitivity for detecting the IgG antibodies in people 17 days after the beginning of symptoms [14].

### 2.4. Statistical Analysis

Categoric variables are presented as frequency and percentages while continuous variables are shown as means +/− standard deviation (SD) and as a median and interquartile range. The antibody levels are expressed as a geometric mean +/− standard error (SE). The comparison of the continuous values between the groups was carried out with a nonparametric Mann–Whitney or Kruskal–Wallis test. Spearman’s Rank Order Correlation test investigated the association between age and SARS-CoV-2 antibody response. A univariate general linear (two-way ANOVA model) was used to evaluate the association between the absolute variation between M1 and M2 of the IgG-trimeric antibodies after the second dose (D2), and sex, age, BMI and the serum IgG antibody levels, starting 2–4 weeks after D2. A multivariable linear regression model to account for possible confounding factors (age, sex, and BMI) associated with the immune response to the anti-COVID-19 vaccine during M1 and M2 was used. The null hypothesis was refused with *p* < 0.05, and statistical analysis was carried out with the IBM SPSS 26.0 statistical package.

## 3. Results

### 3.1. Study 1 (S1)

#### 3.1.1. IgG Values

290 health care workers that received the second dose of the COVID-19 vaccine and met the established criteria of non-previous or current COVID-19 disease participated in the study. A total of 86 of them (30%) were male and 204 (70%) were female. The mean age of participants was 50.0 ± 9.84 years; the mean age of men was 52.0 ± 11.0 years, and the mean age of women was 49.5 ± 9.26 years. This difference was marginally insignificant. The mean Body Mass Index (BMI) of participants was 26.05 ± 4.84; the mean BMI of men was 29.34 ± 4.46, and the mean BMI of women was 25.4 ± 4.69. The difference in BMI between men and women was statistically significant (*p* < 0.0001).The demographic (age and sex), anthropometric (BMI) and clinical characteristics (SARS-CoV2 IgG response) of participants in S1 are presented in Table 1, and their detailed age distribution is depicted in Figure 1.

All participants (290 out of 290, 100%) developed sufficient IgG titers in serum, higher than the cut off value of 50 AU/mL (mean 12,741 ± 597 AU/mL). In total, 242 out of 290 participants (83%) developed IgG titers ranging in the normal range (50–25,000 AU/mL), while 48 out of 290 participants (17%) developed very high IgG levels (>25,000 AU/mL). Most participants (189/290, 65%) developed IgG titers ranging from 5000 to 20,000 AU/mL. Only 1 out of 290 participants (0.35%) developed low IgG levels (<1000 AU/mL) but none were below 800 UA/mL.

#### 3.1.2. Relation of Sex and Age to IgG Values

Women developed higher IgG titers than men (13,759 ± 725 AU/mL versus 10,203 ± 1010 AU/mL, respectively, *p* = 0.007, Figure 2). The IgG titers were inversely related to age in both sexes (*p* = 0.001, Figure 3). When IgG titers were stratified in age groups, a statistical difference (*p* = 0.001) was observed between the youngest and the oldest age group (<30 and >60) and between the youngest and the group aged 51–60 years (*p* = 0.006) (Figure 4). The difference between the age groups appeared to be of practical significance (partial eta squared 0.054). Nevertheless, the mean IgG titers of men and women in each age group did not differ significantly (*p* = 0.181). There was no significant difference in the effect of age on IgG titers for both sexes (two-way ANOVA model, interaction effect of age*sex *p* = 0.866).

#### 3.1.3. Relation of BMI to IgG Values

When participants were stratified in 4 groups according to their BMI (<18.5, 18.5–24.9, 25–29.9, > 30, i.e., underweight, normal weight, overweight and obese, respectively), there was a small tendency of IgG titers to be inversely related to BMI, but this change was not significant (*p* > 0.05, Figure 5).

### 3.2. Study 2 (S2)

180 health care workers participated in Study 2, in which IgG levels were measured six months after the first measurement that took place 2–4 weeks after the second dose of the BNT162b2 COVID-19 vaccine. In total, 61 participants (34%) were male and 119 participants (66%) were female. The mean age of the participants was 51.5 ± 8.0 years.

The demographic, anthropometric and clinical characteristics of participants included in S2 are reported in Table 2.

#### 3.2.1. IgG Values

Six months after the first measurement (M1), the IgG titers of both men and women were decreased dramatically to values less than 5% of the initial: the mean IgG value was 564 ± 57.8 AU/mL (95% lower than the initial), with an IgG value of 596 ± 78.7 AU/mL for women and 502 ± 74 AU/mL for men (Figure 6). This difference in IgG values between males and females was not statistically significant (*p* = 0.256).

In spite of the dramatic decrease in IgG values six months after the first measurement, 176 out of 180 participants (98%) had sufficient serum IgG titers, higher than the cut off value of 50 AU/mL; only 4 participants (2%) had IgG titers below the cut off value of 50 AU/mL.

#### 3.2.2. Difference in IgG Values According to Age and Sex

The mean IgG decrease for all participants in S2 was 11,180 ± 449 AU/ML with a minimum value of 965.9 AU/ML and maximum value of 25,274 AU/ML. Women had a mean decrease of 11,641 ± 561 AU/ML (min 1225, max 25,274), while for men, the mean decrease was 10,474 ± 737 AU/ML (min 966, max 22,917). However, this difference was not statistically significant, *p* = 0.139.

The decrease in IgG titers appeared to be inversely related to age (*p* = 0.003, Figure 7). In particular, when the univariate general linear analysis (two-way ANOVA model) was performed, the IgG decrease was statistically significant between the age groups 41–50 and 51–60 years old (*p* = 0.008). However, men and women did not differ in terms of their IgG decrease (two-way ANOVA model, interaction effect of age*sex, *p* = 0.068).

#### 3.2.3. IgG Values According to BMI

Participants were divided into four groups according to their BMI: <18.5, 18.5–24.9, 25–29.9, and >30 (underweight, normal weight, overweight and obese, respectively), and a one-way ANOVA model was performed. IgG titers exhibited a trend to be inversely related to BMI, but this relation was not significant (*p* = 0.812) (Figure 8).

### 3.3. Multivariate Regression Analysis

A multivariate regression analysis was used to predict the immune response to the anti-COVID-19 vaccine in relation to sex, age and BMI in our study population in S1. The examined parameters (age, sex and BMI) statistically significantly explained 9% of the variance in SARS-CoV-2 IgG titers (R2 = 0.09, *p* < 0.0005). Only age and sex added statistical significance to the prediction, *p* < 0.05. Of these two variables, age made the largest unique contribution (age β value = −0.275 versus sex β value = 0.141) (Table 3). In particular, nearly 28% of the observed variability in antibody response was due tο age and 14% due to sex. The effect of BMI was limited and not significant: only 6% (β value = 0.062, *p* = 0.352).

The equation that describes the above associations is as follows:Predicted immune response to the anti-COVID-19 vaccine = 20,091 + (3138 × sex) − (284 × age) + (129.8 × BMI).

## 4. Discussion

The first COVID-19 case in Greece was confirmed on 26 February 2020, when a 38-year-old woman from Thessaloniki (the biggest city in Northern Greece), who had recently visited Northern Italy, was confirmed to be infected. Following the confirmation of the second and third cases in Greece on 27 February 2020, the first restriction measures against COVID-19 transmission were established. Restriction measures, including school closure and the suspension of cultural events, were taken locally at first, and culminated in restrictions on all non-essential movement throughout the country, starting from 6 a.m. on 23 March 2020 [15,16].

The restriction measures applied in Greece proved to be effective in slowing the spread of the disease and kept the number of deaths among the lowest in Europe [17], but had a high economic cost and a high burden on quality of life throughout the country.

Fortunately, the first vaccinations began in Greece on 27 December 2020, just 10 months after the first COVID-19 case. The first COVID-19 vaccine available in the country was the Pfizer-BioNTech BNT162b2 COVID-19 vaccine, a monovalent mRNA vaccine encoding the spike protein of the original SARS-CoV-2. The first target group of the population for vaccination were health care workers throughout all hospitals in the country.

In the General Hospital of Katerini, the vaccination of health care workers began on 5 January 2021, and was implemented by the end of March, according to the availability of the BNT162b2 COVID-19 vaccine, which was the only COVID-19 vaccine available when vaccination was started in our hospital.

The successful immunization of health care professionals against COVID-19 is of great importance worldwide, as it may provide an effective protection wall against the transmission of COVID-19 among vulnerable hospitalized patients.

Our work confirms the high protection against SARS-CoV-2 offered by the BNT162b2 COVID-19 vaccine, as the two-dose immunization proved to be successful for all participants in the study, and the initial IgG titers, measured two to four weeks after a second dose of the vaccine, were very high. These results are in accordance with other findings [18,19,20].

We are not sure about the exact role of anti-SARS-CoV-2-Spike IgG antibodies in the actual immunization against SARS-CoV-2. Nevertheless, their serum titer is an easy, simple, and measurable index of immunity against SARS-CoV-2 after COVID 19 vaccination.

We must note that apart from the role of T and B cells in immunization against COVID-19, host innate immune response plays an even more important role, as the activation of T and B lymphocytes is connected to the primary antiviral action of the innate immune system that is expressed by cytokine and interferon secretion, antigen presentation and the killing of infected cells by Natural Killer (NK) cells. In fact, the first line of defense against the virus is represented by innate immunity and the action of NK cells, neutrophils and monocytes/macrophages. This innate immunity is necessary in determining the activation of an efficient and specific acquired immune response [21,22,23,24].

In our study, women developed higher IgG titers than men, and this difference was statistically significant. Although there are some studies supporting the contrary observation that men developed higher IgG titers than women [25] or that women developed insignificantly higher IgG titers than men [26], most studies found that females tend to produce higher titers than males [27,28,29,30,31]. This finding is in accordance with the generally accepted view about sex-based immunological differences that contributed to higher morbidity and mortality in men due to COVID-19 infection as well [9,10]. Although the prevalence of COVID-19 is equal between men and women, the risk for worse outcome and death is higher in men than women, irrespective of age [8,32].

According to our findings, the IgG titers were inversely related to age in both sexes. This finding is almost universal and common in other studies, too [25,27,28,29,31]. Older people develop lower IgG titers, and this is consistent with higher vulnerability to COVID-19 and other infections as well [1,2]. Especially during the COVID-19 pandemic, old age was considered one of the main risks for high morbidity and mortality due to COVID-19 disease [6,7].

Although age and sex attracted attention in almost every research work studying humoral immunization after COVID-19 vaccination, BMI was not equally explored, in spite of the fact that obesity has already been considered among the main risk factors for morbidity and mortality due to COVID-19 from the beginning of the pandemic [1,2]. In our study, we observed a small tendency of IgG levels to fall by increasing BMI, but this change was not significant. By dividing the sample into four groups according to BMI values of underweight, normal weight, overweight and obese, we observed a higher longitudinal decrease in the antibody levels when BMI was increased; we observed a steeper slope between groups of lower BMI, with the slope decreasing from group to group with increasing BMI.

In our study, we found high levels of serum anti-SARS-CoV-2-Spike IgG antibodies in the first measurement that was performed two to four weeks after the second dose of COVID-19 vaccine. Nevertheless, six months after the first measurement, the initially high IgG titer was reduced to just 5% of the initial titer. Although the exact meaning of this decrease cannot be evaluated, the results of our study support the importance of the third dose of COVID-19 vaccine. A longitudinal decrease in IgG levels has been observed in other studies as well [12,33,34].

To our knowledge, this is one of the few studies that measured COVID-19 IgG after COVID-19 vaccination in Greece, and the only study measuring COVID-19 IgG antibodies after vaccination in the city and the broader area of Katerini. Additionally, it was one of the few studies that explored the role of BMI in the development of IgG antibodies post-COVID-19 vaccination in Greece.

A strong point of our work was the use of multivariate regression analysis for a quantitative confirmation of the role of age, sex and BMI in IgG levels after COVID-19 vaccination. However, the equation that described this association explained with statistical significance only 9% of the variance in SARS-CoV-2 IgG titers in the studied population.

The limitations of our study include the lack of a high number of participants older than 65 years, and the unavailability of molecular or serological testing for COVID-19 infection prior to IgG measurement. In fact, at the time of the study, there was an urgent need for evaluating COVID-19 infection, as well as the effectiveness of the newly developed vaccines against COVID-19, by measuring the anti-spike IgG antibodies as indices related to acquired immunity due to vaccination. Ιn order to save enough test kits for clinical use (that was a serious priority at that time), we decided to conduct this study without prior laboratory testing that confirmed previous COVID-19 disease. However, all the infected health workers of our hospital had been tested with the appropriate molecular or serological tests that confirmed current COVID-19 disease, in case of disease or suspicion of disease; so, we could exclude from the study either participants that were infected during the study or participants that had been infected before the study. In this way, data that could affect the results were not included in the study.

## 5. Conclusions

In conclusion, the results of this study show that after two doses of BNT162b2 vaccine, women developed higher IgG titers than men, and the IgG titers were lower in older people of both sexes. Six months after the first measurement, the IgG titers of both men and women fell dramatically to values less than 5% of the initial, mitigating the differences observed in the first measurement between both the two sexes and age groups. Apart from age and sex, BMI did not play a significant role in IgG levels after COVID-19 vaccination. On the contrary, the role of age and sex in IgG levels after COVID-19 vaccination was confirmed and determined using multivariate regression analysis.

## Figures and Tables

**Figure 1 microorganisms-11-01279-f001:**
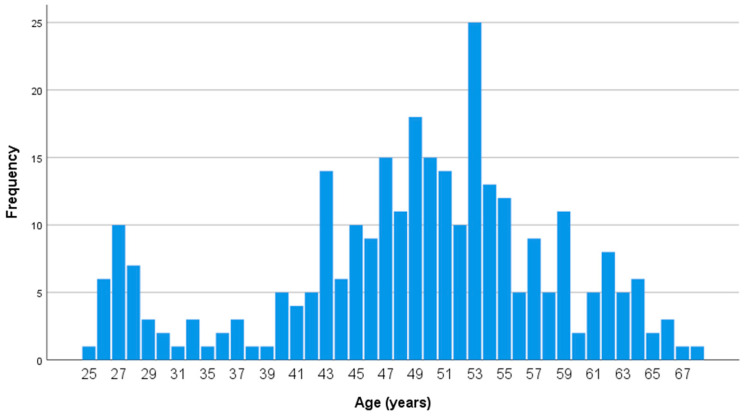
Age distribution of participants in S1 (*n* = 290, 86 men and 204 women). SARS-CoV-2 IgG antibody titers were measured 2–4 weeks after the second dose of the vaccine.

**Figure 2 microorganisms-11-01279-f002:**
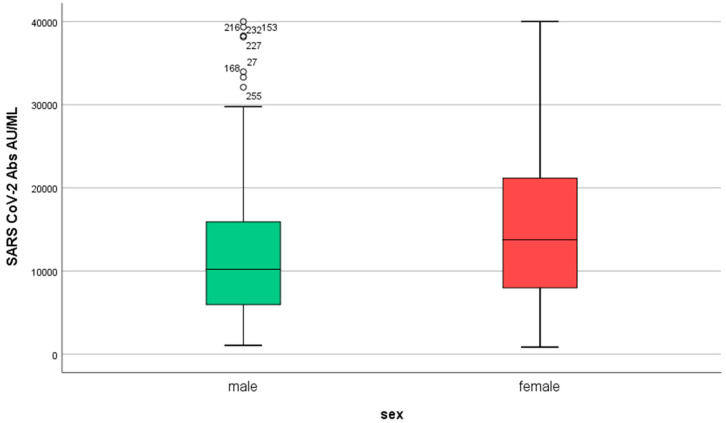
IgG serum levels in males and females (*n* = 290, 86 men and 204 women) in S1. SARS-CoV-2 IgG antibody titers were measured 2–4 weeks after the second dose of the vaccine.

**Figure 3 microorganisms-11-01279-f003:**
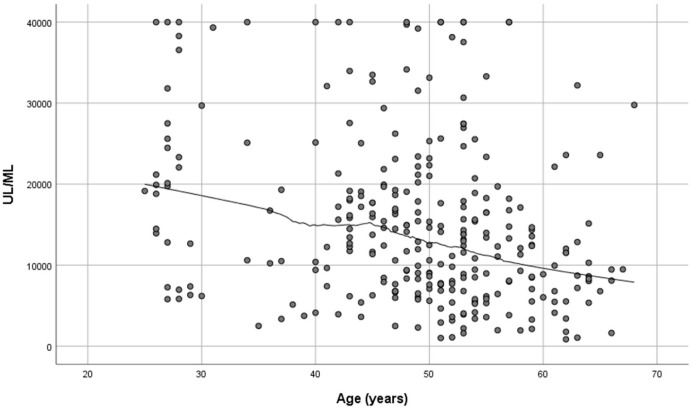
SARS-CoV-2 antibody titers at 2–4 weeks post-vaccination plotted against age (*n* = 290). Spearman’s r = −0.276, *p* = 0.01.

**Figure 4 microorganisms-11-01279-f004:**
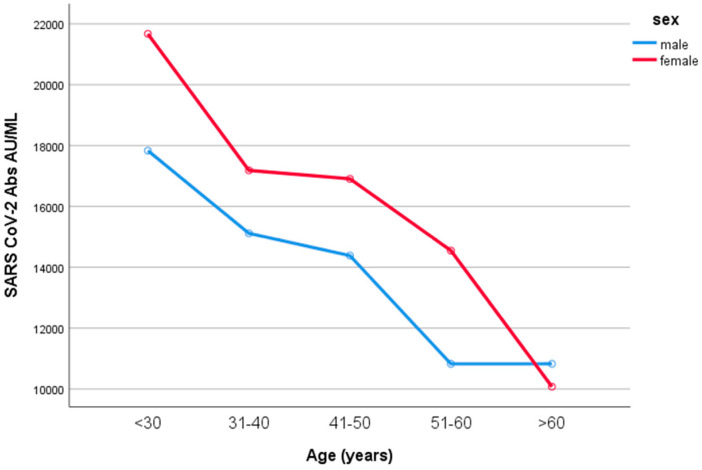
IgG values according to age in males and females. When IgG titers were stratified in age groups, a statistical difference (*p* = 0.001) was observed between the youngest and the oldest age group (<30 and >60) and between the youngest and the group 51–60 years old (*p* = 0.006). Nevertheless, the mean IgG titers of men and women in each age group did not differ significantly (*p* = 0.181).

**Figure 5 microorganisms-11-01279-f005:**
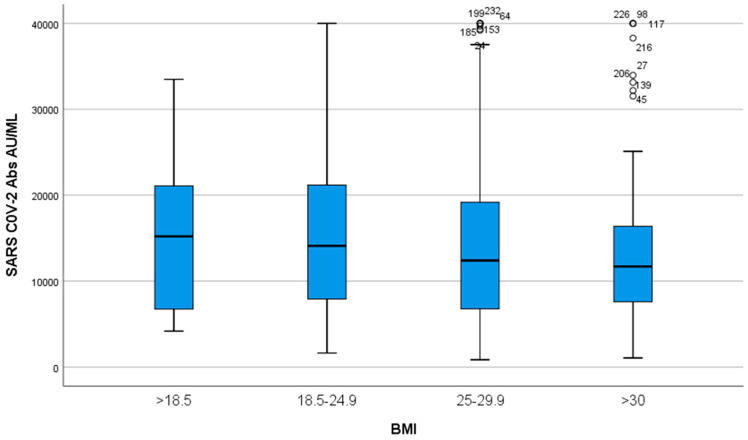
IgG titers according to BMI. Participants were divided into four groups according to their BMI: <18.5, 18.5–24.9, 25–29.9, >30. A small tendency of IgG titers to be inversely related to BMI was observed (*p* > 0.05).

**Figure 6 microorganisms-11-01279-f006:**
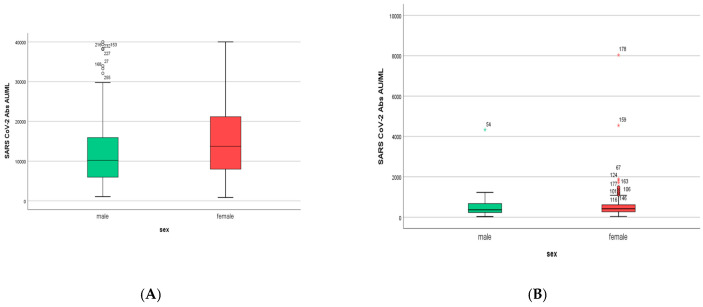
(**A**). IgG titers in men and women included in S1. Women developed higher IgG titers than men (13,759 ± 725 AU/mL versus 10,203 ± 1010 AU/mL, respectively, *p* = 0.007). (**B**). IgG titers in men and women included in S2. Women had higher IgG titers than men (596 ± 858 AU/mL versus 500 ± 858 AU/mL, respectively, *p* = 0.256).

**Figure 7 microorganisms-11-01279-f007:**
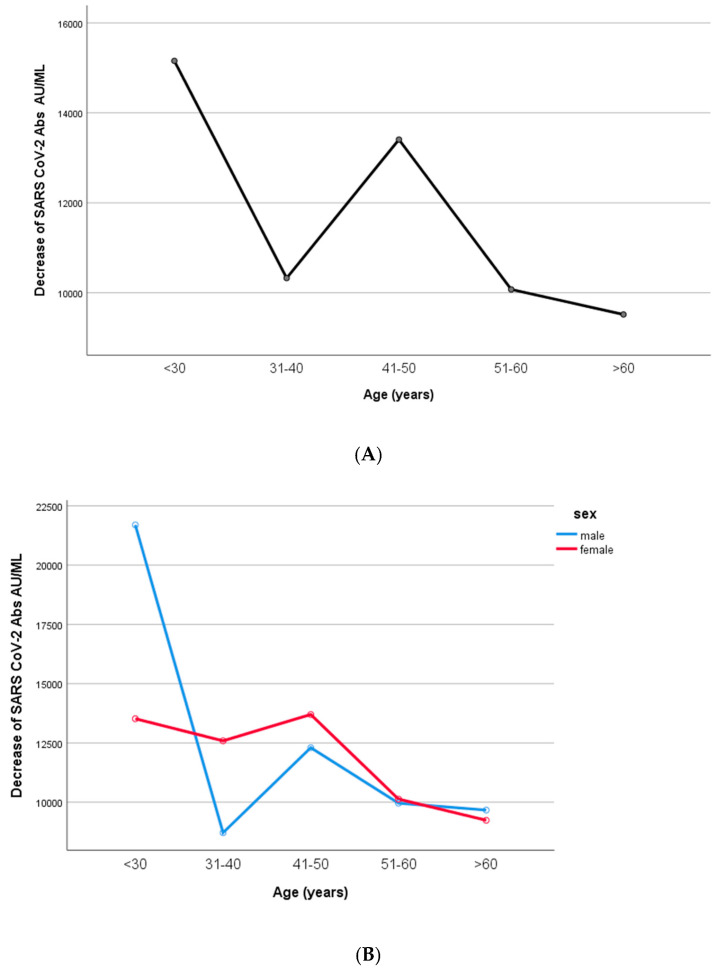
Differences in SARS-CoV-2 antibodies according to age groups. (**A**). All participants. (**B**). Men and women. The IgG decrease was inversely related to age (*p* = 0.003). In particular, IgG decrease was statistically significant between the age groups 41–50 and 51–60 years old (*p* = 0.008). Men and women did not differ in terms of their IgG decrease (two-way ANOVA model, interaction effect of age*sex, *p* = 0.068).

**Figure 8 microorganisms-11-01279-f008:**
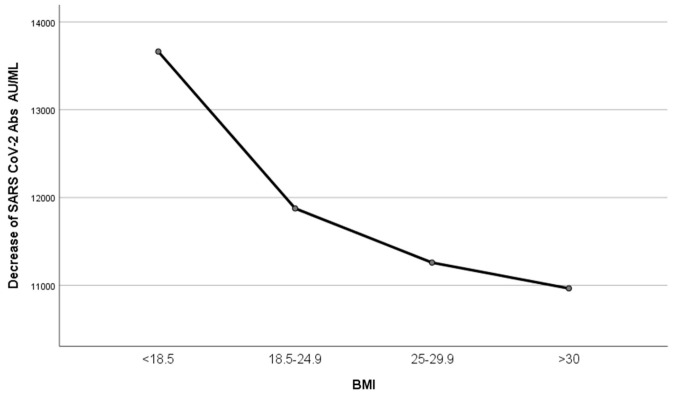
Decrease in IgG titers according to BMI. Participants were divided into four groups according to their BMI: <18.5, 18.5–24.9, 25–29.9, and >30, and one-way ANOVA was performed. IgG titers exhibited a trend to be inversely related to BMI (*p* = 0.812).

**Table 1 microorganisms-11-01279-t001:** Demographic, anthropometric and clinical characteristics of participants included in S1.

	Total Sample	Men	Women	*p* (Men-Women)
Participants	290	86 (30%)	204 (70%)	
Mean age (±SD) (min–max)	50.0 (±9.84) (25–68)	52.0 (±11.0) (25–68)	49.5 (±9.26) (26–66)	0.05
Mean SARS-CoV2 IgG value (±SE) (min–max)	12,741 ± 597 (min 852.5; max = 40,000)	10,203 ± 1010 (min 1052; max = 40,000)	13,759 ± 725 (min 852.5; max = 40,000)	0.007
Mean BMI (±SD) (min–max)	26.05 (±4.84) (18.0–42.1)	29.34 (±4.46) (18.9–42.1)	25.4 (±4.69) (18.0–41.0)	<0.0001

S1: Study 1 (SARS-CoV-2 IgG antibody titers were measured 2–4 weeks after the second dose of the vaccine), BMI: Body Mass Index.

**Table 2 microorganisms-11-01279-t002:** Demographic, anthropometric and clinical characteristics of participants included in S2.

	Total Sample	Men	Women	*p* (Men-Women)
Participants	180	61 (34%)	119 (66%)	
Mean age (±SD) (min–max)	51.5 (±8.0) (26–67)	52.4 (±9.1) (28–67)	50.0 (±7.3) (26–66)	0.035
Mean SARS-CoV2 IgG value (±SE) (min–max)	564 ± 57.8 (min 27; max = 8034)	502 ± 74.0 (min 27; max = 4335)	596 ± 78.7 (min 40; max = 8034)	0.256
Mean BMI (±SD) (min–max)	26.6 (±5.0) (18.0–42.1)	29.6 (±4.66) (18.4–42.1)	25.5 (±4.86) (18.0–40.4)	0.000

S2: Study 2 (6 months after M1, 9 August to 31 August 2021). M1: the day of the 1st measurement of IgG, 26 January 2021 to 26 February 2021; the 1st measurement of IgG was performed 2–4 weeks after the second dose of COVID-19 vaccine.

**Table 3 microorganisms-11-01279-t003:** Multivariate regression analysis performed by using sex, age and BMI as independent variables associated with the immune response to the anti-COVID-19 vaccine in all participants included in S1.

	β Value	Standard Error of β Value	*p* Level
Sex	0.141	1400	0.026
Age	−0.275	65.6	0.000
BMI	0.062	139	0.352

## Data Availability

Data supporting the reported results are kept by the authors and are available on demand, but they cannot be shared publicly because there is no consent provided by participants on the use of their confidential data.

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
