# Peer review of "Age, Sex and BMI Relations with Anti-SARS-CoV-2-Spike IgG Antibodies after BNT162b2 COVID-19 Vaccine in Health Care Workers in Northern Greece"

_microorganisms, 2023, doi:10.3390/microorganisms11051279_

Round 1

Reviewer 1 Report

In a population of greek healthcare workers, the study confirms previous findings that the antibody response to SARS-CoV-2 inversely correlates with age, is higher in females than in males and potentially lower in  individuals with overweight. The study design is straight forward and the results sound. I have two minor suggestions: 

1. To exclude previous SARS-CoV-2 infection sera should have also been tested for anti-N SARS-CoV-2 antibodies. 

2. The number of figures should be reduced by eliminating Figure 1 and by putting several graphs into one Figure. Figure 2 has low quality/resolution. 

Without checking extensively, I stumbled over a few spelling and grammar mistakes. Please check. 

Author Response

Dear Reviewer,

Thank you very much for your comments and suggestions.

We have made all appropriate changes in our manuscript, according to your suggestions, related either to our study and manuscript or to spelling and grammar mistakes. You will find detailed information on these changes in the attached file: “Detailed answer to Reviewer 1”.

The Assigned Editor has asked us to add some more details in our manuscript; according to her suggestion, we have added new text and details in the following sections of the manuscript: Materials and Methods, Results, and Discussion.

You may also find a new version of our manuscript, with changes according to the suggestions of the Reviewers and the Editors. All changes - even the most minor changes in spelling - are presented in red color.

With kind regards,

On behalf of all Authors

Paraskevi Papaioannidou

Reviewer 2 Report

The manuscript entitled “Age, sex and BMI relations with anti-SARS-CoV-2-Spike IgG antibodies after BNT162b2 COVID-19 vaccine in health care workers in Northern Greece” of Papaioannidou P et al. study the correlation between anti-SARS-CoV-2-Spike IgG antibodies and age, sex and BMI after BNT162b2 COVID-19 vaccine in health care workers in Northern Greece. The manuscript is well written, organized and of great interest. I suggest this work for publication after minor revision:

- the resolution of Figure 2 is too low. Please furnish an higher-solved image.

Author Response

Dear Reviewer,

Thank you very much for your comments and suggestion.

According to your suggestion, we have substituted Figure 2 with another, clearer version.

The Assigned Editor has asked us to add some more details in our manuscript; according to her suggestion, we have added new text and details in the following sections of the manuscript: Materials and Methods, Results, and Discussion.

You may find a new version of our manuscript, with changes according to the suggestions of the Reviewers and the Editors. All changes, even the most minor changes in spelling, are presented in red color.

With kind regards,

On behalf of all Authors 

Paraskevi Papaioannidou